# Research and Application of a Quantitative Prediction Method for Sandstone Thickness in a Zone with Dense Well Pattern Development Based on an Objective Function

Cao Li [1,2], Yongzhuo Wang [2], Yongbing Zhou [2], Xiaodong Fan [2], Jianfei Zhan [2], Guosong Chen [2,*] and Zongbao Liu [2]

[1] School of Earth Science, Northeast Petroleum University, Daqing 163318, China
[2] Research Institute of Exploration and Development, Daqing Oilfield Company Limited, PetroChina, Daqing 163712, China
* Correspondence: dydlcgs@163.com

**Abstract:** The seismic amplitude along a layer in a section can reveal lateral reservoir changes and is one of the important means of reservoir prediction, but it is often difficult to establish the quantitative relationship between the seismic amplitude and sandstone thickness in wells in blocks with denser development. The use of a lower-amplitude slice for precisely quantitatively predicting the sandstone thickness, based on the goal of obtaining data on layers in the development zone with a dense well pattern, is accurate and has a valuable advantage. The method of formation slice optimization based on an objective function is studied to improve the ability of reservoir characterization by seismic attributes. This method has been applied to the reservoir prediction of the G $I_2$ sedimentary unit in the ZQX block of the SRT oilfield and has achieved good results.

**Keywords:** stratal slice; sandstone thickness; quantitative prediction

## 1. Introduction

The value of seismic sedimentology in studying sedimentary sand bodies (especially thin sand bodies) and stratigraphic lithologic reservoirs in complex sedimentary sequences is being recognized by many geologists [1,2]. Dahm and Graebner [3] first identified the paleogeomorphological features of meandering channels in seismic time–amplitude slices. Subsequently, Horizon Slice software was developed by some geophysical software companies (Landmark, Hong Kong, China, GeoQuest, etc.) and is widely used by the petroleum industry. In the 1990s, Zeng [4] proposed and discussed the concept of stratal slices and how to build them. He published in the journal *Geophysics*. The concept of seismic sedimentology is proposed for the first time. Seismic sedimentology is the seismic study of sedimentary rocks and their sedimentary processes, and the study is based on mapping the lithologic geomorphology (or units or zones) via jointly studying the seismic lithology and seismic geomorphology [5–7]. Moreover, Posamentier [8] proposed the concept of seismic geomorphology and used geomorphological imaging features of slices to carry out sedimentological interpretation. The following two basic principles of seismic sedimentology research have been generally accepted [9,10]: (1) the width of general sedimentary systems is much larger than the thickness [11]; (2) laterally, it is possible to identify geological bodies that are difficult to identify by seismic vertical resolution through seismic transverse resolution (i.e., the Fresnel zone, where the vertical and transverse resolutions are the same in seismic data with fully realized 3D migration). A stratal slice is one of the key techniques in seismic sedimentology. This method uses two isochronous sedimentary interfaces as reference layers and interpolates a series of seismic amplitude slices. These seismic amplitude slices are called stratal slices. The essence of sedimentology research using seismic stratigraphy slicing technology is to study the reservoir lithology

and deposition by using single-amplitude point information. Because the different types of river geomorphology are all banded, it is easy to identify fluvial thin layer sand bodies by stratal slices [2]. Meandering river sand bodies with different degrees of bending have the characteristics of "long shoelace strips", especially the seismic responses of beach-edge deposits. Braided river sand bodies have the characteristics of "wide banding" contiguous distribution. However, the reticular river bodies have obvious geomorphologic features of a "curved reticular strip" [2].

In addition, sedimentology/seismic geomorphology originated from the study of sedimentary systems of marine clastic rocks. Based on the high precision sequence stratigraphic framework, it has made remarkable progress in the analysis and quantitative characterization of the undercut valley, low fan, low delta, transgressive shoreline sand body, high barrier sand bar, lagoon and delta sand body, and the bending characteristics of channel/channel in different sequences or system domains [12–14]. In addition, the combination of seismic geomorphology (coherent slice), seismic geometric attribute extraction, RGB (Red–Green–Blue) fusion and plane imaging techniques can characterize carbonate karst paleo-geomorphology and carbonate/evaporite and clastic mixed deposits [15,16]. Above all, quantitative parameters and configurations of sand bodies of different sedimentary types can be studied by seismic sedimentology combined with a variety of geophysical techniques, such as determining the camber of channel system, channel width, river bend zone width, river bend arch height and the relationship between these parameters [17,18]. The results show that the amplitude class attribute has the best correlation with the thickness of the sand body, and the maximum peak amplitude is the best. Compared with the original seismic attribute, the maximum peak amplitude attribute obtained by frequency division fusion can better describe the boundary of sand bodies of different origin and different thickness. It can quantitatively characterize the thickness, distribution and shape of sand bodies as well as analyze and predict the reservoir quality [18].

In this paper, because the main target layer of the Daqing Changyuan oilfield is a continental fluvial delta sedimentary system deposit, with longitudinal sand and mudstone interaction and rapid transverse phase transformation, it is very difficult to quantitatively predict the sand body. Due to the limitation of seismic vertical resolution, it is usually difficult for seismic interpretation horizons to accurately correspond to small layers (thicknesses within the tunable thickness range). It is necessary to carry out further research on the optimization and optimization technology of stratal slices to improve the correlation between the stratum slice amplitude and sandstone thickness. By establishing the functional relationship between the seismic amplitude and sandstone thickness, the quantitative prediction of sandstone thickness under the control of reliable seismic amplitude is established to improve the accuracy of reservoir prediction.

## 2. Geological Background

The Daqing Changyuan oilfield is located in the secondary structural zone of Daqing Changyuan in the central depression of the Songliao Basin. It is a large anticlinal sandstone oil and gas field. The study area is located in the Saertu anticline in the northern Daqing Changyuan oilfield (Figure 1). The target strata are the lower part of the Nenjiang Formation, the upper part of the Yaojia Formation and the upper part of the Qingshankou Formation, and the main oil reservoirs, namely, Saertu, Tuohua and Gaotaizi, are developed (Figure 2).

The Saertu, Putaohua and Gaotaizi oil reservoirs in the Songliao Basin were part of the fluvial delta sedimentary system of a large continental lacustrine basin. The upper depression supersequence is composed of the Yaojia Formation and Nenjiang Formation, the bottom interface is Sq19 (88 Ma), and the top interface is Sq4 (72 Ma), which is the unconformity between the Nenjiang Formation and Sifangtai Formation. The lake rose to the highest level again at the end of the first member of the Nenjiang Formation deposition and the beginning of the second member of the Nenjiang Formation deposition, during which the oil shale deposits became a dense segment of the supersequence and formed

a set of source rocks in the Songliao Basin. The sedimentary area reached its maximum during this period, and its scope was far beyond that of the present basin. It can be further divided into six third-order sequences, namely, Sq19, Sq20, Sq21, Sq22, Sq23, and Sq24 from bottom to top, which are equivalent to the Nenjiang Formation of the Yaojia Association (Figure 2). The G $I_2$ sedimentary unit in the ZQX block of the SRT oilfield is the main target of this study.

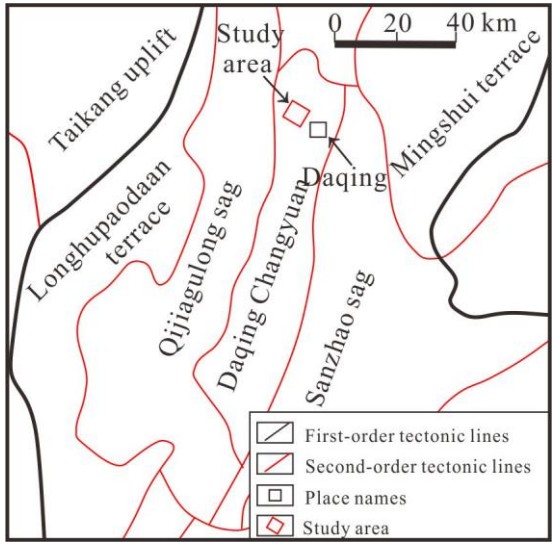

**Figure 1.** Regional location map of the Daqing Changyuan oilfield.

| System | Series | Formation | Member | Resrvoirs | Oil layers | Super sequence | | Sea level change |
|---|---|---|---|---|---|---|---|---|
| Cretaceous | Upper cretaceous | Neijiang | Five | Heidimiao (H) | | Upper | Sq24 | Rise / Decline |
| | | | Four | | | | Sq23 | |
| | | | Three | | | | Sq22 | |
| | | | Two | | | | | |
| | | | One | | | | Sq21 | |
| | | yaojia | Two-Three | Saertu (S) | SO | | | |
| | | | | | S I | | | |
| | | | | | S II | | Sq20 | |
| | | | | | S III | | | |
| | | | One | Putaohua (P) | P I | | Sq19 | |
| | | | | | P II | | | |
| | | Qingshankou | Three | Gaotaozi (G) | G I | Lower | Sq18 | |
| | | | | | G II | | Sq17 | |
| | | | | | G III | | Sq16 | |
| | | | Two | | G IV | | | |
| | | | One | | | | Sq15 | |

**Figure 2.** Characteristics of the stratigraphic framework of the Daqing Changyuan oilfield.

## 3. Reservoir Prediction Method Based on Objective Function

Widess [19] noted that when the formation thickness is less than 1/4 of the seismic wavelength, the reflection amplitude is proportional to the formation thickness. The seismic response characteristics of thin interbedded layers are complicated due to the influence of the lithologic structure and depth, and the variation in the reflection amplitude is obvious. As a result, stratigraphic slices cannot be directly used to predict the thickness of reservoir sandstone.

Based on the wave theory method and the actual data of the Daqing research area, forward modeling of thin interbedded wedges is carried out. The results show a good correlation between the seismic amplitude and the sandstone thickness in the Daqing Changyuan study area, and the sandstone thickness can be well evaluated quantitatively by the variation in the seismic amplitude. Therefore, in this paper, Pearson correlation analysis theory is used to automatically select the best matching well–seismic stratum slice, and the

functional relationship between the stratum slice amplitude and sandstone thickness can be obtained from the well point statistical law. Finally, by using this objective function to search the appropriate amplitude value in the hour window, the correlation between the amplitude and the sandstone thickness can be further improved. As a result, the quantitative prediction of sandstone reservoirs can be achieved by the amplitude information.

The objective function method for the quantitative prediction of sandstone reservoirs can be divided into three parts: (1) Analyze the relationship between the seismic amplitude along the layer and seismic response time window and obtain the functional relationship between the amplitude along the layer and reservoir sandstone thickness (objective function). (2) Then, the amplitude values consistent with the data model are locked in the seismic data volume of the short-time window encryption sampling and corrected to screen the best amplitude values. (3) Finally, the sandstone reservoir thickness is predicted by the regression calculation method [20].

### 3.1. Seismic Geological Correspondence Analysis

Synthetic seismic records are a common method of seismic geological horizon calibration and a bridge between high-resolution well logging information and regional seismic information. The following principles should be followed in the preparation of synthetic seismic records in areas with dense well patterns [21,22]. First, the characteristics of the standard seismic reflection layer and the patterns of vertical and horizontal distribution should be determined according to the existing knowledge of the research area. The second principle is to correct the logging data and to fit the missing density curve. Third, it is necessary to optimize the seismic theory wavelet or extract the sidetrack wavelet. The fourth principle is to compare the similarity or correlation between the synthetic seismographic trace and the seismographic trace and establish the corresponding relationship between seismogeological layers by adjusting the depth relationship automatically or manually. The location information and reservoir information can be accurately demarcated on the seismic profile by finely calibrating multiple wells.

### 3.2. Constructing Stratal Slices

Based on the fine calibration results of multiple wells, the seismic response characteristics of relatively stable and approximately isochronous sedimentary interfaces in sediments can be analyzed, and the lateral stability of these features can be studied. Generally, the seismic interpreter selects a stable interface with stable seismic characteristics that is easy to track laterally and has similar deposition as those of the standard layers above and below the target layer. Based on the two standard layers, the time and thickness between the two standard layers are calculated. To meet the demand of thin sand body development as much as possible, the number of stratigraphic sections is usually taken as the number of sections identified via an earthquake, which is greater than the maximum stratal thickness. In this way, the seismic section set can be constructed to ensure that all seismic sample information can be sliced. In addition, the detailed process of seismic slice picking up is available in reference [23].

### 3.3. Automatic Optimization of Stratal Slices

In seismic sedimentology studies, a stratigraphic section that is consistent with the sedimentary pattern of the target layer is usually selected by seismological or geological experts to study seismic deposition or seismic geomorphology. This optimization process usually requires researchers to perform a manual comparative analysis based on regional geological knowledge or well data and is often referred to as the "expert selection method". This method greatly depends on the selection, adjustment and human experience of stratigraphic slice color standards, and the work efficiency is also low, which has a certain degree of human subjectivity. To solve this problem, Pearson's correlation analysis theory is used.

Pearson's product difference correlation method is a correlation calculation method proposed by British statistician Pearson at the beginning of the 20th century. It is a com-

mon method to calculate correlation coefficients and is also the most common and basic method to reveal the direction and extent of the linear correlation of two variables. The Pearson correlation coefficient between two variables is defined as the quotient of covariance and standard deviation between two variables (Equation (1) is visible in this article). The Pearson correlation analysis method was applied in the geophysical exploration field in 1980, mainly for the analysis of well seismic parameter relationships, and many successful examples were obtained, but it was not applied to the optimization of seismic sedimentology slices.

In this paper, the author developed an automatic optimization method for stratigraphic sections and programmed the corresponding computer program to realize the automatic optimization of stratigraphic sections with the best coincidence between the well shock in the target stratum. This method can not only quickly and automatically select the best stratigraphic section but also determine the range of sliding time windows with the same seismic response, which can provide a reference for the further optimization of amplitude sections in the range of small time windows.

This is an example equation:

$$r = \frac{\sum xy - \frac{\sum x \sum y}{N}}{\sqrt{\left(\sum x^2 - \frac{(\sum x)^2}{N}\right)\left(\sum y^2 - \frac{(\sum y)^2}{N}\right)}} \tag{1}$$

where $r$ is the correlation coefficient; $x$ is the seismic property of the well point; $y$ is the thickness of the wellpoint sandstone; and $N$ is the number of wells drilled.

### 3.4. Objective Function Establishment and Amplitude Slice Optimization

According to the range of the automatic optimal combination and sliding window of the slice, the sandstone thickness and seismic amplitude data of all wells were extracted based on the corresponding relationship between the coordinate position of the target layer and the seismic surface element. The intersection diagram of the seismic amplitude and sandstone thickness was established, and the functional relationship between them was established by means of unary linear regression analysis. Generally, the data points in this intersection diagram are scattered in the area with a dense well pattern, and the range of sandstone thickness corresponding to the single-amplitude data is very wide. Therefore, it is necessary to carry out further optimization processing of amplitude slices to improve the seismic correlation of the well.

In this paper, the objective function of this area in Formula (2) is established based on the initial correlation between the seismic amplitude of the slice along the layer and the thickness of sandstone.

The establishment method of the objective function is based on the sandstone thickness and seismic amplitude data of all wells, taking the sandstone thickness data as the independent variable and the seismic amplitude data as the dependent variable, and using the unary linear regression method to fit the regression coefficient, so as to obtain the linear relationship formula between the seismic amplitude and sandstone thickness, which is the objective function of this region and this horizon.

Based on this function, the dominant amplitude data with a good corresponding relationship with the reservoir are determined, and then, the seismic slice is optimized.

The optimization method of seismic slice is to optimize the seismic slice by automatically adjusting the seismic horizon. To do this,

(1) One option is to calculate the seismic reflection time position of the nearest target amplitude value at each single well point (hereinafter represented by TWT1);

(2) The second is to calculate the seismic reflection time difference ($\triangle$TWT) between TWT1 and the seismic reflection time of the current horizon (hereinafter represented by TWT2), i.e., $\triangle$TWT = TWT1 − TWT2;

(3) Thirdly, the method is applied to calculate the △TWT at all well points, and the reflection time correction grid (G(△TWT)) in the study area is drawn with the data.

(4) Fourth, the same grid parameters were used to plot the seismic reflection time of the current horizon, G(TWT2);

(5) Fifth, calculate the corrected time grid, G(△TWT1) = G(TWT2) + G(△TWT);

(6) Sixth, the time grid data after correction is converted into a seismic reflection horizon, and the optimized seismic slices are extracted.

$$Amp = a * H + b \tag{2}$$

where *Amp* is the amplitude of the slice; *H* is the sand thickness; and *a* and *b* are coefficients.

## 4. Examples of Application

Based on the above principles, a series of computer programs were programmed with FORTRAN high-level language, including slicing set construction, stratigraphic slicing optimization, objective function establishment and slicing optimization. This method is applied to the ZQX block of the SRT oilfield.

The top G I oil layer of the ZQX block is the result of delta front deposition, and reservoir lateral variation is rapid. The ZQX area G I upper reservoir group is the result of delta front deposition, and reservoir lateral variation is rapid; making use of seismic lateral resolution for reservoir prediction provides a geological basis. The thickness of sedimentary layer G $I_2$ in the research area ranges from 0 to 12.3 m, the sandstone thickness ranges from 0 to 8.6 m, and the sandstone thickness and $\lambda/4$ (approximately 15 m) in this area theoretically conform to the relation between the amplitude and the thickness of sandstone. Through all the calibrations of the detailed seismic geologic horizon in the well, the G $I_2$ sandstone units in the seismic data mainly show corresponding trough reflection characteristics. The calligraphist slice set was constructed by a computer program, and the slice along the amplitude of the layer with the closest distribution pattern was selected (Figure 3). Through the intersection analysis of the seismic amplitude along the layer at the well point and sandstone thickness data (Figure 4), it was found that the intersection data presented a banded distribution and that a negative correlation existed. Because the intersection of data points is more dispersed, single-amplitude data corresponding to the sandstone thickness range are very widespread; the objective function method and procedures for the regression coefficient of functions a = −18,560 and b = 42,812, respectively, which are set up along the layer in the seismic amplitude slice and have a relationship with the initial sandstone thickness, are based on this function to determine a good advantage of the corresponding relationship with the reservoir amplitude data.

By applying the establishment method of the objective function discussed above and using the unitary linear regression method, the relationship between seismic amplitude and sandstone thickness of the G $I_2$ layer in the ZQX block is fitted, namely formula (2), and the coefficients a and b in Formula (2) are obtained.

On the basis of the objective function, in the hour window of the well (2 ~ + 2—ms), within the scope of the search and conforming to the amplitude of the values, the amplitude value corresponding to the return trip and seismic reflection time difference with the original stratigraphic section is determined. According to the district, all wells (1565, which has a well density of approximately 160/km$^2$) have an advantage over the original stratigraphic section of the seismic reflection amplitude difference of the G $I_2$ stratigraphic section in the study area of the seismic horizon adjustment, yielding an amplitude section advantage (Figure 5).

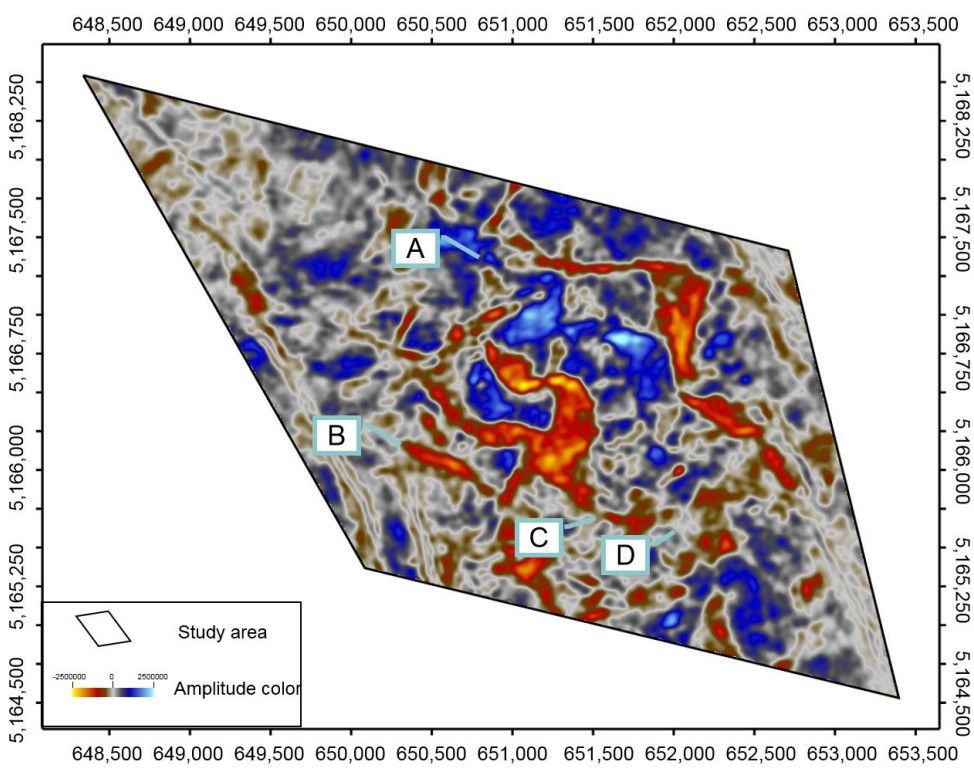

**Figure 3.** Amplitude slice along layer G $I_2$. A, B, C and D represent four different typical amplitude variation locations.

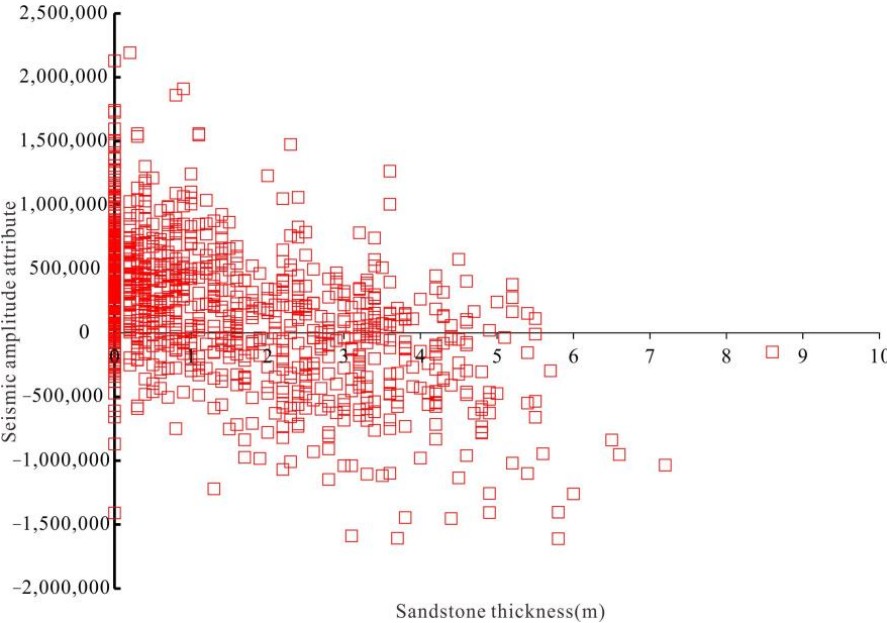

**Figure 4.** Cross-plot of the amplitude and sandstone thickness of layer G $I_2$.

The intersection analysis of the dominant amplitude at the well point and the thickness of sandstone shows that the intersection data of the two parameters reveal an obvious banded distribution (Figure 6). The correlation between seismic amplitude and sandstone thickness is better, the correlation coefficient increases from 0.51 to 0.75, and the linear correlation is more obvious.

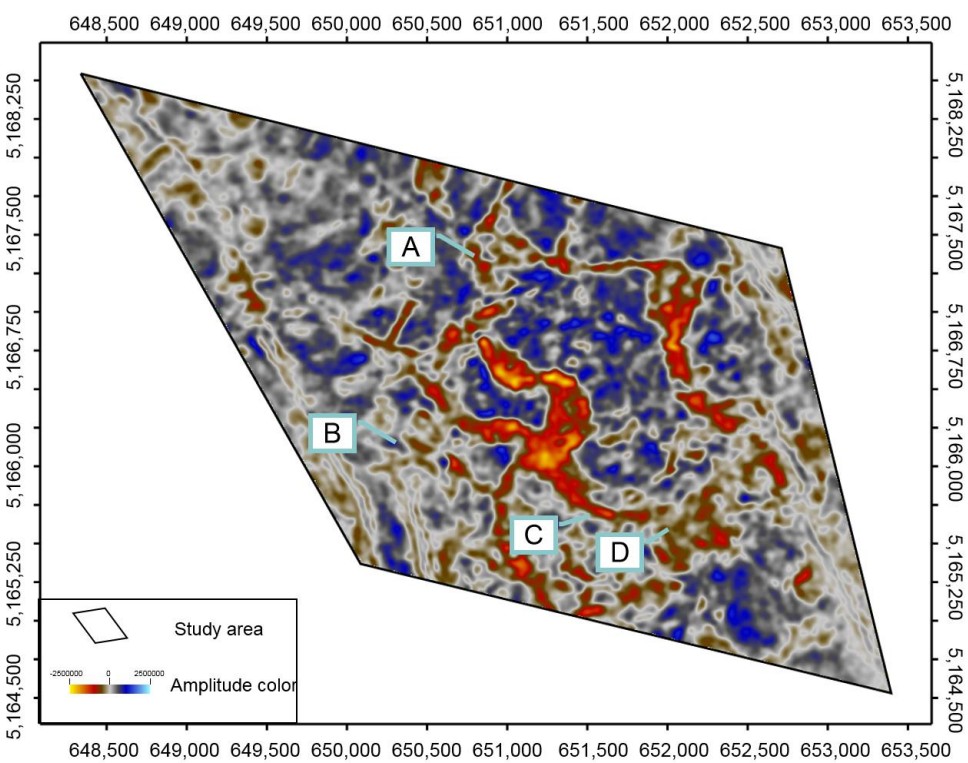

**Figure 5.** Dominant amplitude of layer G $I_2$. A, B, C and D represent four different typical amplitude variation locations.

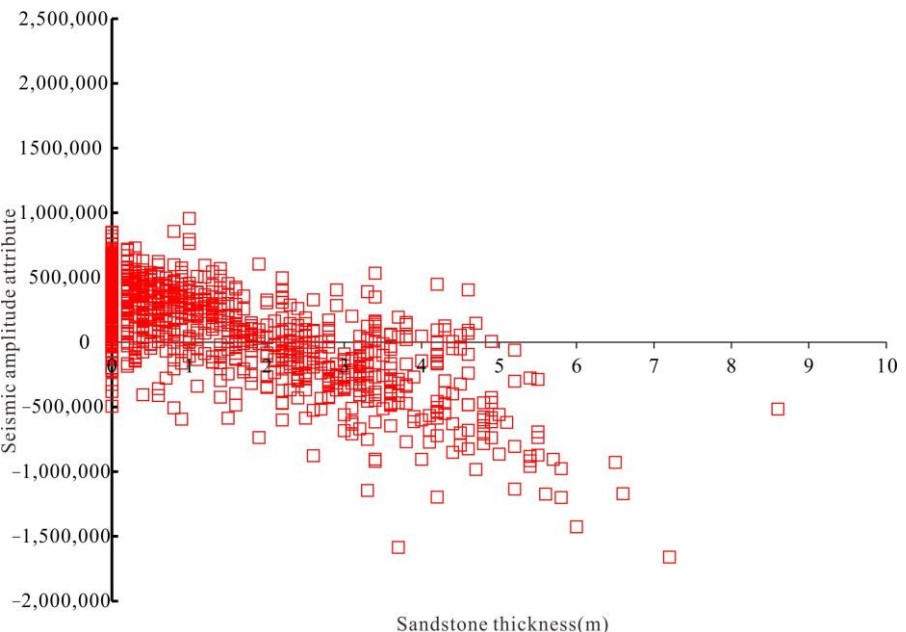

**Figure 6.** Relationship between the dominant amplitude and sandstone thickness in layer G $I_2$.

The data points are concentrated, and the positive correlation is more obvious. The dominant amplitude slice (Figure 5) is consistent with the preferred amplitude slice along the layer (Figure 3), which can show the distribution of the sand body. Many changes have occurred locally, and these changes are similar to the shape of the sand body thickness on the contour map [24,25].

The contour map of sandstone thickness was drawn using well data in the development zone (Figure 7). Due to the high density of wells in this area, which reached 160/km$^2$,

the contour map could well reflect the plane distribution of sand bodies. In Figures 3 and 5, negative amplitude (warm tone) indicates relatively developed sandstone, while positive amplitude (cool tone) indicates no development of sandstone. According to the analysis in Figure 7, the dominant amplitude slice (Figure 5), compared with the preferred along-bed amplitude slice (Figure 3), maintains the main bands that can reflect the distribution trend of sand bodies, with many changes in local details, which are more consistent with the thickness of sandstone at the well point.

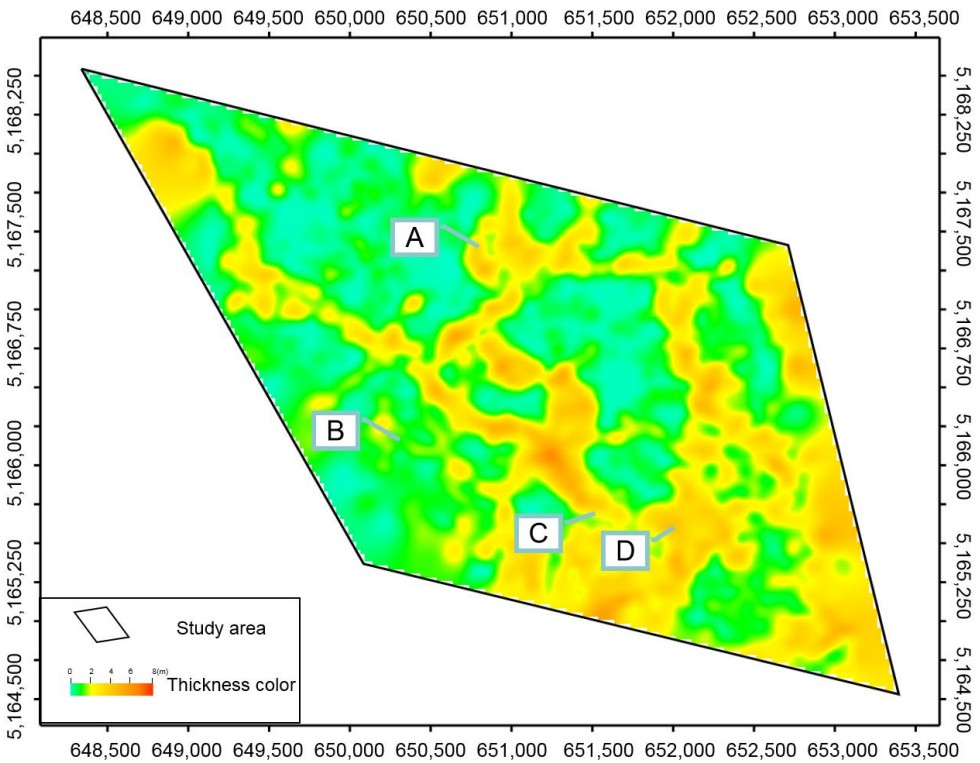

**Figure 7.** Sandstone thickness prediction map of G I2. A, B, C and D are four different typical thickness variation locations of sandstone, corresponding to amplitude variation locations.

For example, the thickness contour map of the sandstone at position A shows that the thickness of sandstone is relatively large, and the dominant amplitude slice is negative amplitude, which is consistent with the thickness of the sandstone, while the amplitude slice along the layer is positive amplitude, which is inconsistent with the thickness of the sandstone.

The thickness contour map of the sandstone at position B shows that the thickness of the sandstone is small, and the dominant amplitude slice is positive amplitude, which is consistent with the thickness of the sandstone, while the amplitude slice along the layer is negative amplitude, which is inconsistent with the thickness of the sandstone.

The thickness contour map of sandstone at position C shows that the thickness of the sandstone is relatively large, and the dominant amplitude slice is negative amplitude, which is consistent with the thickness of the sandstone, while the amplitude slice along the bed is positive amplitude, which is inconsistent with the thickness of the sandstone.

The contour map of sandstone thickness at position D shows that the sandstone thickness is relatively large. The dominant amplitude slice is negative amplitude, which is consistent with the thickness of the sandstone, while the amplitude slice along the layer is positive amplitude, which is inconsistent with the thickness of the sandstone.

Importantly, the correlation between the dominant amplitude and the thickness of the sandstone has significantly improved, and the two different parameters can more adequately show the same geological information at the well point, so the prediction results

of the horizontal reservoir based on the slice information of the dominant amplitude are more reliable.

According to the relationship between the sandstone thickness and dominant amplitude, the prediction map of the sandstone thickness can be calculated via the slice data of dominant amplitudes by using unary linear regression (Figure 8). Under the condition that the error of the absolute thickness is less than 1 m, the coincidence rate of well seismic matching can reach 85.6%, and the prediction result is good. The variation between wells is obvious. For example, in the middle and eastern parts of the study area, many banded sand bodies have the same shape as narrow channel sand bodies along the preferred amplitude slice. In the middle of the working area, the belt where the main channel sand bodies are present shows more detailed changes, such as the change in the thickness of the sand body inside the channel sand body and the local muddy interval. Dynamic and static analyses and verification of multiple wells and multilevel groups are adopted to confirm that these changes are real and reliable. The changes in these details indicate that in the zone in which the tight well pattern developed, there are still many sand body changes in the main channel belt that cannot be controlled by well data, and narrow channels with small widths and half-well distances between wells may be present. On this basis, well vibration combined with the anatomy of these geological phenomena is of great significance for oilfield development. The changes in these reservoirs and the corresponding geological knowledge provide a reference for the development of the ZQX block of the SRT oilfield to explore the potential, adjust the well position, and play a practical role in the development of the oilfield.

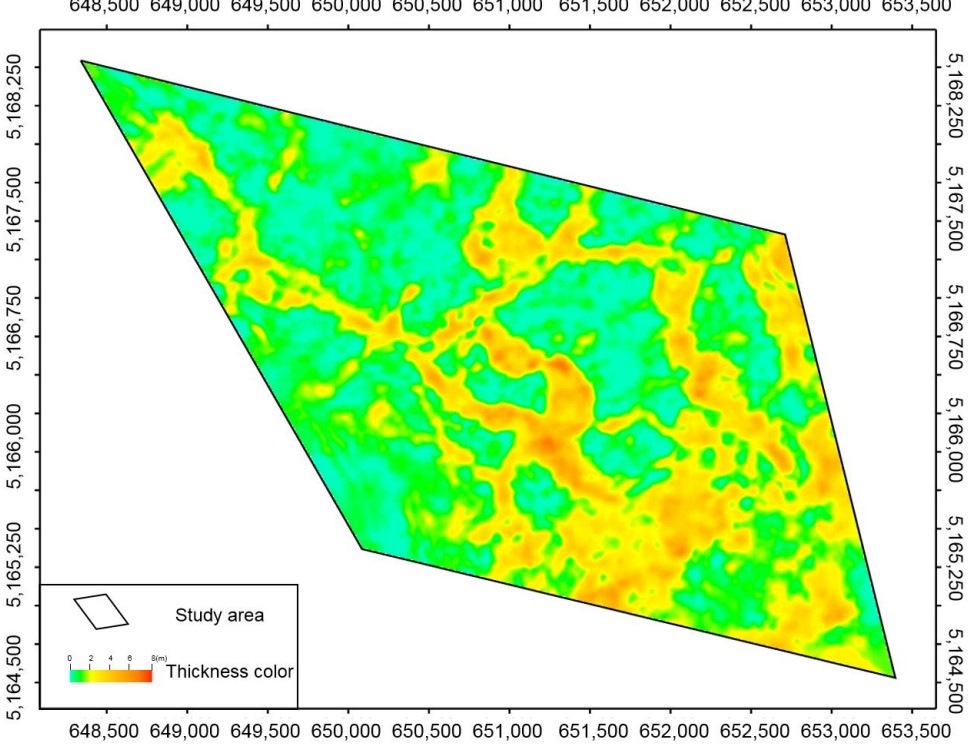

**Figure 8.** Sand thickness prediction map of G $I_2$.

In addition, this method has been applied in many blocks of the Daqing Changyuan oilfield, and it has improved the accuracy of reservoir prediction. At present, it has not been applied in other oil fields, but from the analysis of the principle, this method can be applied in the well pattern of relatively uniform well pattern combination reservoir prediction work, and it is expected to achieve good application results.

This method is suitable for development zones but not for early exploration blocks with low well density. The premise of application of this method is that the reservoir thickness

is less than one-quarter of the seismic wavelength. According to Widess' theory [19], there is a linear relationship between seismic amplitude and reservoir thickness.

## 5. Conclusions

(1) The stratigraphic slice corresponding to the sedimentary unit-level reservoirs that are required for geological research can be selected by automatic section selection. Although the stratigraphic section can reveal the characteristics of planar sedimentary facies by showing the variation pattern of sandstone thickness, the correlation between sandstone thickness and amplitude is poor under the condition of a dense well pattern, and it is difficult to establish the correlation between them.

(2) For the reservoir in which sand and mudstone longitudinally interact, the inter-well reservoir can be quantitatively evaluated by establishing the relationship between the sandstone thickness and amplitude and adjusting the effective pickup horizon in a short window.

(3) For the zone in which a dense well pattern developed, the channel still contains a variety of sand body changes, which are not constrained by well data, and even narrow channels with widths less than half the well spacing may be present between wells. The combination of seismic and well logging technology can effectively constrain this problem to a certain extent and provide scientific guidance for oilfield development.

**Author Contributions:** Conceptualization, C.L.; methodology, C.L. and Y.W.; investigation, G.C. and Z.L.; resources, Y.Z., X.F. and J.Z.; data curation, Y.Z., X.F. and J.Z.; writing—original draft preparation, C.L.; writing—review and editing, C.L. and G.C.; supervision, Y.Z., X.F. and J.Z.; project administration, Y.Z., X.F. and J.Z. All authors have read and agreed to the published version of the manuscript.

**Funding:** This research was funded by [National Natural Science Foundation of China] grant number [42172161].

**Institutional Review Board Statement:** This article does not cover that. Therefore, relevant statements can be excluded.

**Informed Consent Statement:** Informed consent was obtained from all subjects involved in the study.

**Data Availability Statement:** Not applicable.

**Conflicts of Interest:** The authors declare no conflict of interest.

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
