# Peer review of "Research and Application of a Quantitative Prediction Method for Sandstone Thickness in a Zone with Dense Well Pattern Development Based on an Objective Function"

_processes, doi:10.3390/pr11010281_

Round 1

Reviewer 1 Report (New Reviewer)

See Attached Document

Author Response

In my opinion, the article probes an interesting problem. Nevertheless, before I recommend this work for publication, the authors should carefully and fully address my comments shown below and improve the literature studies: 

  • The authors need to provide more details on the objective function establishment and amplitude slice optimization. It is important to discuss this in more details to allow others use/repeat this work. 

The corresponding modification has been made ,see details on lines 213-234: “The establishment method of the objective function is based on the sandstone thickness and seismic amplitude data of all Wells, taking the sandstone thickness data as the independent variable and the seismic amplitude data as the dependent variable, and using the unitary linear regression method to fit the regression coefficient, so as to obtain the linear relationship formula between the seismic amplitude and sandstone thickness, which is the objective function of this region and this horizon.Based on this function, the dominant amplitude data with a good corresponding relationship with the reservoir are determined, and then the seismic slice is optimized.

The optimization method of seismic slice is to optimize the seismic slice by automatically adjusting the seismic horizon. To do this,(1)One is to calculate the seismic reflection time position of the nearest target amplitude value at each single well point (hereinafter represented by TWT1);(2)The second is to calculate the seismic reflection time difference (â–³TWT) between TWT1 and the seismic reflection time of the current horizon (hereinafter represented by TWT2), i.e. â–³TWT= TWT1-TWT2;(3)Thirdly, the method is applied to calculate the â–³TWT at all well points, and the reflection time correction grid (G(â–³TWT)) in the study area is drawn with the data.(4)Fourth, the same grid parameters were used to plot the seismic reflection time of the current horizon, G(TWT2);(5)Fifth, calculate the corrected time grid, G(â–³TWT1)= G(TWT2)+ G(â–³TWT);(6)Sixth, the time grid data after correction is converted into seismic reflection horizon, and the optimized seismic slices are extracted.”

  • Can the proposed method be applied to other oil fields rather than Daqing Changyuan oilfield? Please state the limitation(s) of the proposed approach/methodology

  The corresponding modification has been made ,see details on lines 351-359: In addition, this method has been applied in many blocks of Daqing Changyuan oilfield, and has improved the accuracy of reservoir prediction. At present, it has not been applied in other oil fields, but from the analysis of principle, this method can be applied in the well pattern of relatively uniform well pattern combination reservoir prediction work, and it is expected to achieve good application results.

This method is suitable for development zones, but not for early exploration blocks with low well density. The premise of application of this method is that the reservoir thickness is less than a quarter of the seismic wavelength. According to widess theory [19], there is a linear relationship between seismic amplitude and reservoir thickness.

  • What are the main assumptions considered in constructing stratal slices as well as automatic optimization of stratal slices? The authors should include all the assumptions made in the main text as it helps other researchers in the field to advance this work. 

The corresponding modification has been made ,see details on lines 351-359: The premise of application of this method is that the reservoir thickness is less than a quarter of the seismic wavelength. According to widess theory [19], there is a linear relationship between seismic amplitude and reservoir thickness.

(4) Please provide more in-depth analysis with regards to  cross plot of the amplitude and sandstone thickness of layer G I2. Additionally, compare this with the literature reports.  

The corresponding modification has been made ,see details on lines 293-323: The contour map of sandstone thickness was drawn using well data in the development zone (Figure. 7). Due to the high density of Wells in this area, which reached 160 /km2, the contour map could well reflect the plane distribution of sand bodies. In the Figure. 5 and Figure. 3, negative amplitude (warm tone) indicates relatively developed sandstone, while positive amplitude (cool tone) indicates no development of sandstone. According to the analysis in the Figure. 7, the dominant amplitude slice (Figure. 5), compared with the preferred along-bed amplitude slice (Figure. 3), maintains the main bands that can reflect the distribution trend of sand bodies, with many changes in local details, which are more consistent with the thickness of sandstone at the well point.

For example, the thickness contour map of sandstone at position A shows that the thickness of sandstone is relatively large, and the dominant amplitude slice is negative amplitude, consistent with the thickness of sandstone, while the amplitude slice along the layer is positive amplitude, inconsistent with the thickness of sandstone.

The thickness contour map of the sandstone at position B shows that the thickness of the sandstone is small, and the dominant amplitude slice is positive amplitude, consistent with the thickness of the sandstone, while the amplitude slice along the layer is negative amplitude, inconsistent with the thickness of the sandstone.

The thickness contour map of sandstone at position C shows that the thickness of sandstone is relatively large, and the dominant amplitude slice is negative amplitude, consistent with the thickness of sandstone, while the amplitude slice along the bed is positive amplitude, inconsistent with the thickness of sandstone.

The contour map of sandstone thickness at position D shows that the sandstone thickness is relatively large. The dominant amplitude slice is negative amplitude, consistent with the thickness of the sandstone, while the amplitude slice along the layer is positive amplitude, inconsistent with the thickness of the sandstone

Importantly, the correlation between the dominant amplitude and the thickness of sandstone has significantly improved, and the two different parameters can more adequately show the same geological information at the well point, so the prediction results of the horizontal reservoir based on the slice information of the dominant amplitude are more reliable.

  • Why did not the authors evaluate the porosity of the oilfield? Is it possible to consider this for the future work? 

The relevant research on oilfield porosity will be further considered in the follow up study.

  • The authors need to highlight the importance of this work and how this is different than other studies.

The corresponding modification has been made ,see details on lines 57-76: “In addition, sedimentology/seismic geomorphology originated from the study of sedimentary systems of Marine clastic rocks. Based on the high precision sequence stratigraphic framework, it has made remarkable progress in the analysis and quantitative characterization of the undercut valley, low fan, low delta, transgressive shoreline sand body, high barrier sand bar, lagoon and delta sand body, and the bending characteristics of channel/channel in different sequences or system domains [12-14]. Also, the combination of seismic geomorphology (coherent slice), seismic geometric attribute extraction, RGB (Red-Green-Blue) fusion and plane imaging techniques can characterize carbonate karst paleo-geomorphology and carbonate/ evaporite and clastic mixed deposits [15-16]. Above all, quantitative parameters and configurations of sand bodies of different sedimentary types can be studied by seismic sedimentology combined with a variety of geophysical techniques, such as determining the camber of channel system, channel width, river bend zone width, river bend arch height and the relationship between these parameters [17-18]. The results show that the amplitude class attribute has the best correlation with the thickness of sand body, and the maximum peak amplitude is the best. Compared with the original seismic attribute, the maximum peak amplitude attribute obtained by frequency division fusion can better describe the boundary of sand body of different origin and different thickness. It can quantitatively characterize the thickness, distribution and shape of sand body, and analyze and predict the reservoir quality [18].”

(7) Improve the quality of all figures including both x-axis and y-axis. The numbers cannot be seen easily. The corresponding modification has been made ,see details in this paper.

(8) The manuscript requires proofreading, there are several grammatical as well as typographical issues in the text.

 The corresponding modification has been made ,see details in this paper.

(9) The literature review needs to be improved. The authors can acknowledge the work of different groups: The corresponding modification has been made ,see details in this paper.

(i) https://doi.org/10.1139/cjc-2021-0248

(ii) https://doi.org/10.1016/j.marpetgeo.2022.106034

(ii) https://doi.org/10.1021/acs.energyfuels.1c00717

Reviewer 2 Report (New Reviewer)

In my opinion, the article probes an interesting problem. Nevertheless, before I recommend this work for publication, the authors should carefully and fully address my comments shown below and improve the literature studies: 

(1) The authors need to provide more details on the objective function establishment and amplitude slice optimization. It is important to discuss this in more details to allow others use/repeat this work. 

(2) Can the proposed method be applied to other oil fields rather than Daqing Changyuan oilfield? Please state the limitation(s) of the proposed approach/methodology . 

(3) What are the main assumptions considered in constructing stratal slices as well as automatic optimization of stratal slices? The authors should include all the assumptions made in the main text as it helps other researchers in the field to advance this work. 

(4) Please provide more in-depth analysis with regards to  cross plot of the amplitude and sandstone thickness of layer G I2. Additionally, compare this with the literature reports. 

(5) Why did not the authors evaluate the porosity of the oilfield? Is it possible to consider this for the future work? 

(6) The authors need to highlight the importance of this work and how this is different than other studies.

(7) Improve the quality of all figures including both x-axis and y-axis. The numbers cannot be seen easily. 

(8) The manuscript requires proofreading, there are several grammatical as well as typographical issues in the text. 

(9) The literature review needs to be improved. The authors can acknowledge the work of different groups:

(i) https://doi.org/10.1139/cjc-2021-0248

(ii) https://doi.org/10.1016/j.marpetgeo.2022.106034

(ii) https://doi.org/10.1021/acs.energyfuels.1c00717

Author Response

The content of the manuscript was of high quality. However, there were certain aspect of the

manuscript that requires attention by the authors.

  1. Firstly, I found that the manuscript lack of support (references). For example, the following

statements require support/evidence by literaturea. On page 7 lines 231 – 232, “The dominant amplitude slice (Figure 5) is consistent with

the preferred amplitude slice along the layer (Figure 3), which can show the distribution

of the sand body”

  1. On page 7, lines 233 – 234, “Many changes have occurred locally, and these changes

are similar to the shape of the sand body thickness on the contour map”

The corresponding modification has been made ,see details on lines 288-292: “The data points are concentrated, and the positive correlation is more obvious. The dominant amplitude slice (Figure 5) is consistent with the preferred amplitude slice along the layer (Figure 3), which can show the distribution of the sand body. Many changes have occurred locally, and these changes are similar to the shape of the sand body thickness on the contour map [24-25]”

  1. Additionally, the references on this manuscript need to be greatly improved. The references

were limited and also centered to just one research group. This is a major concern.

The corresponding modification has been made ,see details in the section : introduction

  1. On page 6 lines, 210 – 212, the authors stated “Because the intersection of data points is

more dispersed, single-amplitude data corresponding to the sandstone thickness range are

very widespread; the objective function method and procedures for the regression

coefficient of functions 18560 and 42812, respectively…”

Please provide more details about these coefficients.

The corresponding modification has been made ,see details on lines 264-267: By applying the establishment method of objective function discussed above and using unitary linear regression method, the relationship between seismic amplitude and sandstone thickness of GI2 layer in ZQX block is fitted, namely formula (2), and the coefficients a and b in Formula (2) are obtained

  1. In section 3 of the manuscript, please provide information regarding Pearson correlation.

The corresponding modification has been made ,see details on lines 178-189: “Pearson's product difference correlation method is a correlation calculation method proposed by British statistician Pearson at the beginning of the 20th century. It is a common method to calculate correlation coefficients and is also the most common and basic method to reveal the direction and extent of the linear correlation of two variables. The Pearson correlation coefficient between two variables is defined as the quotient of covariance and standard deviation between two variables (Equation (1) is visible in this article). The Pearson correlation analysis method was applied in the geophysical exploration field in 1980, mainly for the analysis of well seismic parameter relationships, and many successful examples were obtained, but it was not applied to the optimization of seismic sedimentology slices.”

  1. On page 4, lines 117 – 118, the authors stated “Finally, the sandstone reservoir thickness

is predicted by the regression calculation method”

Please provide information about the regression calculation technique that was used in

this study

The corresponding modification has been made ,see details on lines 216-235.:”The establishment method of the objective function is based on the sandstone thickness and seismic amplitude data of all Wells, taking the sandstone thickness data as the independent variable and the seismic amplitude data as the dependent variable, and using the unitary linear regression method to fit the regression coefficient, so as to obtain the linear relationship formula between the seismic amplitude and sandstone thickness, which is the objective function of this region and this horizon.Based on this function, the dominant amplitude data with a good corresponding relationship with the reservoir are determined, and then the seismic slice is optimized.

The optimization method of seismic slice is to optimize the seismic slice by automatically adjusting the seismic horizon. To do this,(1)One is to calculate the seismic reflection time position of the nearest target amplitude value at each single well point (hereinafter represented by TWT1);(2)The second is to calculate the seismic reflection time difference (â–³TWT) between TWT1 and the seismic reflection time of the current horizon (hereinafter represented by TWT2), i.e. â–³TWT= TWT1-TWT2;(3)Thirdly, the method is applied to calculate the â–³TWT at all well points, and the reflection time correction grid (G(â–³TWT)) in the study area is drawn with the data.(4)Fourth, the same grid parameters were used to plot the seismic reflection time of the current horizon, G(TWT2);(5)Fifth, calculate the corrected time grid, G(â–³TWT1)= G(TWT2)+ G(â–³TWT);(6)Sixth, the time grid data after correction is converted into seismic reflection horizon, and the optimized seismic slices are extracted.”

  1. In equation (1) of the manuscript, please provide the difference between x and X and y and

Y.

The corresponding modification has been made ,

  1. English in the manuscript should be reviewed

The corresponding modification has been made ,see details

Round 2

Reviewer 2 Report (New Reviewer)

I see the authors did not provide a response/rebuttal letter to my concerns. I will not recommend this work for publication if my comments are not addressed. Again, here are my main comments shown below: 

(1) The authors need to provide more details on the objective function establishment and amplitude slice optimization. It is important to discuss this in more details to allow others use/repeat this work. 

(2) Can the proposed method be applied to other oil fields rather than Daqing Changyuan oilfield? Please state the limitation(s) of the proposed approach/methodology . 

(3) What are the main assumptions considered in constructing stratal slices as well as automatic optimization of stratal slices? The authors should include all the assumptions made in the main text as it helps other researchers in the field to advance this work. 

(4) Please provide more in-depth analysis with regards to  cross plot of the amplitude and sandstone thickness of layer G I2. Additionally, compare this with the literature reports. 

(5) Why did not the authors evaluate the porosity of the oilfield? Is it possible to consider this for the future work? 

(6) The authors need to highlight the importance of this work and how this is different than other studies.

(7) Improve the quality of all figures including both x-axis and y-axis. The numbers cannot be seen easily. 

(8) The manuscript requires proofreading, there are several grammatical as well as typographical issues in the text. 

(9) The literature review needs to be improved. The authors can acknowledge the work of different groups:

(i) https://doi.org/10.1139/cjc-2021-0248

(ii) https://doi.org/10.1016/j.marpetgeo.2022.106034

(ii) https://doi.org/10.1021/acs.energyfuels.1c00717

Author Response

Modify the description, each reply is marked in red font. Changes in the corresponding manuscript are also marked in blue.

Round 3

Reviewer 2 Report (New Reviewer)

The paper is in a good shape for publication. 

This manuscript is a resubmission of an earlier submission. The following is a list of the peer review reports and author responses from that submission.

Round 1

Reviewer 1 Report

Dear authors,

Thanks for your draft. I have read it and I have several comments that you should address before I can analyze whether to recommend it publication in the journal:

Major comments

1. I am missing the whole structure of the paper, an introduction, review, novelties, problem description, proposed new approach, etc.

2. English should go thru a moderate check, many parts are not quite clear. Besides, you are using too many short sentences, try to build a fluid "story" during the paper, for the easiness of the readership.

3. Abstract should contain more about the results, not only "achieved good results".

4. References: Many are old references, only 3 (2020, 2019, 2018) are relatively recent. Please look for more recent literature. Moreover, I observe 6 (out of 15, this is 40%) references to the same author. No one else publishes about the same topic?

Minor comments

1. There are some lines which are not useful to the paper. For instance, page 1, line 32, "he published on the journal Geophysics". OK, but what is the addition of this line?

2. Add the country where Daqing is located.

3. Add a space between the number and its unit.

4. Add references of the graphs taken from different sources.

5. Why I see Wells with capital letters throughout the whole paper.

6. In certain figures (see e.g. 3) the numbers are not clear, improve the colorbar, add a scale. This applies to several figures.

7. Reference 13 is missing the year

Reviewer 2 Report

Please add these refeckees to improve your quality of intdictuion section

Naseer, M.T., 2020. Seismic attributes and reservoir simulation’application to image the

shallow-marine reservoirs of middle-eocene carbonates, SW Pakistan. J. Petrol. Sci.

Eng. 195, 107711.

Naseer, M.T., 2021. Spectral decomposition’ application for stratigraphic-based quantitative controls on Lower-Cretaceous deltaic systems, Pakistan: significances for hydrocarbon exploration. Mar. Petrol. Geol. 127, 104978.

Reviewer 3 Report

The manuscript (Research and application of quantitative prediction method for sandstone thickness in close well pattern development zone based on objective function) has potential after some modification.

The abstract should be rewritten, in its current form, it is not informative and reflects your work.

in the introduction, the novelty of this work should be emphasized

some recent fluvial sediments references can be cited showing it is important as reservoir rocks ( Facies analysis, diagenesis, and petrophysical controls on the reservoir quality of the low porosity fluvial sandstone of the Nubian formation, east Sirt Basin, Libya: Insights into the role of fractures in fluid migration, fluid flow, and enhancing the permeability of low porous reservoirs. Marine and Petroleum Geology, 105986.)

some recent references might be added ex. ( Seismic attributes and static reservoir simulation applications for imaging the thin-bedded stratigraphic systems of the Lower-Cretaceous Lower Goru fluvial resources, Pakistan. Journal of Asian Earth Sciences, 240, 105409.)

add lat and long for figure 1

the paper format should be restructured as research paper standards, results and discussion.

the authors should elaborate their discussion

present the merits and demerits of this method

compare your method with other methods and researcher's findings

much work should be done to be in acceptable form